Tilianin content and morphological characterization of colchicine-induced autotetraploids in Agastache mexicana

http://orcid.org/0009-0008-2355-0335 Martínez-Aguilar Angélica 1
Villanueva Sánchez Evert 2
Valencia-Díaz Susana 1
Estrada-Soto Samuel E. 3
Napsucialy-Mendivil Selene 4
Barba-Gonzalez Rodrigo 5
Alia-Tejacal Iran 6
http://orcid.org/0000-0003-2367-6217 Arellano-García José de Jesús 1
Villegas Torres Oscar Gabriel 6
Cruz Torres Karla Catalina 3
http://orcid.org/0000-0003-1944-4739 Perea-Arango Irene 1 iperea@uaem.mx
1 Centro de Investigación en Biotecnología, Universidad Autónoma del Estado de Morelos , Cuernavaca, Morelos , Mexico
2 Consejo Nacional de Humanidades, Ciencias y Tecnologías, Laboratorio Nacional de Investigación y Servicio Agroalimentario y Forestal, Universidad Autónoma de Chapingo , Texcoco, Estado de México , Mexico
3 Facultad de Farmacia, Universidad Autónoma del Estado de Morelos , Cuernavaca, Morelos , Mexico
4 Departamento de Biología Molecular de Plantas, Instituto de Biotecnología, Universidad Nacional Autónoma de México , Cuernavaca, Morelos , Mexico
5 Centro de Investigación y Asistencia en Tecnología y Diseño del Estado de Jalisco AC , Guadalajara, Jalisco , Mexico
6 Facultad de Ciencias Agropecuarias, Universidad Autónoma del Estado de Morelos , Cuernavaca, Morelos , Mexico
Okpala Charles
Electronic publication date: 2024 Nov 22
Publication date: 2024
Volume: 12
Electronic Location ID: e18545
Received 2024 Jan 31; Accepted 2024 Oct 28
Copyright: © 2024 Martínez-Aguilar et al.
Copyright year: 2024
Copyright holder: Martínez-Aguilar et al.
License: This is an open access article distributed under the terms of the Creative Commons Attribution License, which permits unrestricted use, distribution, reproduction and adaptation in any medium and for any purpose provided that it is properly attributed. For attribution, the original author(s), title, publication source (PeerJ) and either DOI or URL of the article must be cited.
License URL: https://creativecommons.org/licenses/by/4.0/

Keywords: Polyploidization, Agastache mexicana, Flavonoids, Tilianin, Colchicine, Medicinal plant, Tetraploidy, Secondary metabolites, Chromosome number, Flow cytometry

Funding: Consejo Nacional de Humanidades, Ciencias y Tecnologías, CONAHCyT-Mexico 760148 This work was supported by the Consejo Nacional de Humanidades, Ciencias y Tecnologías, CONAHCyT-Mexico for Ph.D. scholarship 760148. The funders had no role in study design, data collection and analysis, decision to publish, or preparation of the manuscript.

==============================
Background

Agastache mexicana Linton & Epling subsp. mexicana (Lamiaceae) is an aromatic medicinal plant, characterized by a high concentration of tilianin, a flavonoid with therapeutic potential in cardiovascular diseases. In this study, we have explored the use of colchicine to obtain autotetraploid lines of A. mexicana and analyze their morphological characteristics. In addition, we aimed to identify polyploid plants with a high content of tilianin.

Methods

In vitro seedlings at the stage of cotyledon emergence were dipped in colchicine solution at 0.0%, 0.1%, 0.3%, and 0.5% (w/v) for 6, 12, and 24 h. Seedlings were cultured on half-strength basal Murashige and Skoog medium supplemented with 20 g/L sucrose. After 2 months, the shoots from surviving seedlings were excised and grown individually in the same medium to obtain plantlets. The ploidy level of all materials was verified through flow cytometry and chromosome counting before acclimatization and transfer to the greenhouse. The investigated characteristics included length, density and stomatal index, leaf area, chlorophyll content, flower size and color, and tilianin content measured by high-performance liquid chromatography.

Results

The most efficient production of tetraploid in terms of percentage was achieved with 0.1% colchicine for 6 h resulting in no generation of mixoploids. Tetraploid plants had twice the number of chromosomes (2n = 4x = 36) and nearly twice the total DNA content (2.660 ± 0.236 pg) of diploids. Most tetraploid A. mexicana plants showed variations in flower and leaf characteristics compared to the diploid controls. High-performance liquid chromatography analysis showed that tetraploid plants with small leaves produced the greatest amount of tilianin; up to 32.964 ± 0.004 mg/g dry weight (DW), compared to diploid plants with 6.388 ± 0.005 mg/g DW.

Conclusion

In vitro polyploidization using colchicine demonstrates potential for enhancing bioactive constituents of A. mexicana. This approach has proven effective in generating elite tetraploid lines with increased tilianin production.

Introduction

Agastache mexicana Linton & Epling subsp. mexicana (Lamiaceae) is an aromatic plant, native to North America. The plant is an herb that can reach a height of one meter or more. Its morphology is typical of the Lamiaceae, with opposite petiolate leaves with serrated margins, square stem, numerous trichomes, and intensely purple labiate flowers. A. mexicana is widely cultivated in central Mexico for its medicinal and ornamental properties (Palma-Tenango, Sánchez-Fernández & Soto-Hernández, 2021). Due to its low seed viability, this species is commercially propagated through vegetative methods, primarily via rhizome or stem cuttings. As an alternative to these traditional methods, simple and efficient protocols for in vitro seed germination and micropropagation have been developed (Carmona-Castro et al., 2019; Copetta et al., 2023).

The inflorescences, stems and, leaves of A. mexicana have been used in Mexican traditional medicine to treat a range of ailments, including insomnia, anxiety, rheumatism, stomach pain, gastrointestinal disorders, and cardiovascular disease (González-Ramírez et al., 2012; Flores-Flores et al., 2016). The medicinal properties of A. mexicana are predominantly attributed to the presence of terpenes and phenolic compounds, particularly tilianin (acacetin-7-glucoside), the most abundant flavonoid identified in A. mexicana plants, with a concentration of 8 mg/g dry weight (Hernández-Abreu et al., 2011). Tilianin is a bioactive compound derived from plant secondary metabolism that manifests various biological activities beneficial to human health, including neuroprotective, anti-atherogenic, anti-hypertensive, cardioprotective, anti-inflammatory, antioxidant, anti-depressant, and antitumor effects, among others (Akanda et al., 2019; Chen et al., 2024). According to the pharmacological findings, tilianin could lead to the development of new drugs for the treatment of cardiovascular diseases (Khattulanuar et al., 2022; Cruz-Torres et al., 2023; Du et al., 2023). However, similar to other medicinal and aromatic plants, few efforts have been applied to increase the valuable secondary metabolites and ornamental characteristics of A. mexicana.

Artificial polyploidization is one of most remarkable breeding strategies for improving desirables traits in plants. This process involves chromosome doubling through the application of mitotic spindle inhibitors to somatic cells. The result is a range of morphological changes, including variations in the number or size of flowers, fruits, roots, and seeds compared to their diploid counterparts. Additionally, this process is often associated with enhanced chlorophyll content and photosynthetic capacity (Wang et al., 2021), increased tolerance to diverse biotic and abiotic stresses (Tossi et al., 2022), and increased production of bioactive secondary metabolites (Niazian, 2019; Niazian & Nalousi, 2020). This approach has been successfully applied to important medicinal plants such as Thymus vulgaris (Shmeit et al., 2020), Artemisia annua (Lin et al., 2011), Cannabis sativa (Parsons et al., 2019), and Papaver somniferum (Mishra et al., 2010), demonstrating remarkable benefits in both agronomic traits and yield of commercially valuable compounds compared to diploids plants.

The polyploidization can be induced using antimitotic chemicals. Among these, the most commonly used agents are trifluralin, oryzalin, and colchicine. Colchicine, a plant alkaloid, has been successfully used to treat seeds, seedlings in vivo shoot tips, and in vitro explants from several genera of the Lamiaceae family for tetraploidy induction, registering a significant effect on the content and composition of bioactive compounds in species of Agastache (Talebi et al., 2017), Dracocephalum (Yavari, Omidbaigi & Hassani, 2011), Lavandula (Urwin, Horsnell & Moon, 2007; Urwin, 2014), Ocimum (Omidbaigi, Mirzaei & Moghadam, 2010), Salvia (Estaji et al., 2017), and Thymus (Tavan, Mirjalili & Karimzadeh, 2015). In the case of the Agastache genus, few studies have been published, which evaluate the effect of the in vitro polyploidization on the qualitative and quantitative production of bioactive compounds. For instance, in A. foeniculum, the variation in ploidy level, significantly affected the physicochemical and morphological characteristics of the tetraploid plants. The polyploid plants showed an increase in essential oil content and chemical composition, as well as an increased tolerance to salt stress (Talebi et al., 2016, 2017, 2021).

Different methods have been reported for the screening and confirmation of polyploidy in plants. Indirect methods involve assessing various morphological, anatomical, and physiological traits, such as pollen diameter, stomatal size, stomatal density and the number of chloroplasts in guard cells. While these methods are simple, rapid and use simple instruments for detection, they are often inaccurate. In contrast, direct assays, such as flow cytometry and chromosome counting in mitotic cells of root tips, are reliable techniques for accurately determining the ploidy level in plants (Manzoor et al., 2019).

The present study explores the potential of artificial chromosome doubling through colchicine treatment for generating novel A. mexicana genotypes with enhanced tilianin production. The objective was to obtain autotetraploid A. mexicana plants and determine whether the manipulation the ploidy level could produce plant variants with higher tilianin content. Furthermore, the investigation assesses the effect of polyploidization on multiple phenotypic traits across ten autotetraploid lines of A. mexicana, including chlorophyll content and various morphological traits in leaves and flowers.

Materials and Methods

Schematic overview of the experimental program

A schematic overview of the experimental procedure for in vitro polyploidization of Agastache mexicana is presented in Fig. 1. The protocol is initiated with the treatment of seedlings using colchicine, followed by agitation of the treated explants on an orbital shaker before being inoculated in culture media. Putative polyploid shoots (lines) that emerged from treated explants were subsequently subcultured. To confirm the ploidy levels of the resulting plantlets, stomatal characteristic analysis, chromosome counting, and flow cytometry were employed.

Figure 1 Schematic overview of experimental procedure.

Rooted plants were transferred to ex vitro conditions for acclimatization and greenhouse growth. To assess the impact of polyploidization on tilianin accumulation and morphological traits, a comparative study was conducted using ten tetraploid lines and the diploid control. The evaluation encompassed several parameters during the vegetative phase, including tilianin content, chlorophyll levels, and leaf area. Additionally, various floral characteristics were examined.

Polyploid induction

Seeds from Agastache mexicana subsp. mexicana were collected from a wild population in Felipe Neri, Tlanepantla, Morelos, Mexico, at an altitude of 2,823 m above sea level. The plants were authenticated at the HUMO herbarium of the Biodiversity and Conservation Research Center of Universidad Autónoma del Estado de Morelos, and a voucher specimen (No. 35766) was deposited in the same herbarium. Five hundred seeds were surface sterilized using 98% ethanol for 5 min, rinsed with sterile water, and then with 1.5% of commercial sodium hypochlorite (Cloralex®) for 2 min. Subsequently, the seeds were rinsed five times in sterile water and germinated on autoclaved half-strength basal Murashige and Skoog medium (MS 50%, Murashige & Skoog, 1962), containing 50 mg/L myo-inositol and 3.0 g/L Phytagel®. The cultures were maintained at 25 ± 1 °C under a light regime of 16/8 h (light/dark) photoperiod with a light intensity of 2,000 lux.

To observe the effect of the colchicine concentration and exposition time on the survivor rate and polyploidization of A. mexicana subsp. mexicana, we performed an experimental design where 12 seedlings at the stage of cotyledon emergence were dipped in each of the colchicine concentrations 0.0, 0.1, 0.3, and 0.5% (w/v) at three times: 6, 12 and, 24 h (n = 144). Colchicine was sterilized by filtration through a 0.22 μm Millipore syringe filter. Seedlings were incubated with constant shaking at 100 rpm, in the dark at 25 °C to avoid deterioration of the antimitotic agent under light (Eng, Ho & Ling, 2021). Dimethyl sulfoxide (DMSO) was added at a non-toxic concentration of 2% to help the colchicine penetrate through cell walls (Głowacka, Jeżowski & Kaczmarek, 2009; Salma, Kundu & Mandal, 2017). At the end of each treatment, the seedlings were rinsed eight times with sterile water and then transferred to the same medium, supplemented with 20 g/L sucrose. After 2 months, putative polyploid shoots (8–10 cm length) from surviving seedlings were excised and individually cultured into 250 mL glass jars containing 50 mL of MS 50% medium and subcultures, every 20 days. The jars were sealed with aluminum foil caps with tiny holes covered with a piece of 3 M Micropore™ tape that enables reducing humidity in the jars, while increasing gas exchange to minimize the effect caused by hyperhydricity or vitrification (Zarate-Salazar et al., 2020). The cultures were incubated under the same conditions as described previously. Plantlets obtained by in vitro culture of single shoots from colchicine-treated and untreated seedlings were identified as putative polyploid lines (Amx) and diploid control, respectively. After 8 weeks of culture, the stomatal characteristics, such as length, density, and stomatal index, were assessed. Finally, chromosome duplication of in vitro plants was confirmed by chromosome counts in young root tips and flow cytometry of leaf nuclei.

Acclimatization

After 15 months of successive subcultures and confirmation of the ploidy level, the in vitro plantlets of diploid controls and tetraploid lines were transferred to a commercial substrate (peat moss, leaf soil, coconut fiber, tezontle and agrolite) in six-inch plastic pots and grown under greenhouse conditions (25 ± 2 °C and natural photoperiod conditions) with daily irrigation. At least five plants per line were grown, and subsequently, leaf area, chlorophyll content, flower characteristics, and tilianin content were analyzed.

Stomatal characteristics

Five fully expanded leaves were excised at node position three from the shoots of three diploid/or control plants and from three individuals randomly chosen from five selected putative polyploid lines of the in vitro cultures (n = 90, three plants × five leaves × six lines including the control). The abaxial leaf surfaces were coated with a thin layer of nail polish. After 10 min, the dried polish was removed by applying a strip of transparent one-sided adhesive tape. The dry polish samples, along with the adhered sticky tape were mounted permanently on glass microscope slides, and the stomata length, stomata density (number of stomata per visual field, PVF), and stomata index were recorded using a Leica DM500 optical microscope (Leica Microsystems, Wetzlar, Germany). Images were analyzed using image J software (https://imagej.net). The fields of view were located in the middle portion of leaf lamina, and three fields of vision were investigated for each leaf. The stomatal index was determined from the formula: SI = [S/(S + E)] × 100, where S is the number of stomata in the microscopic field and E is the number of epidermal cells per unit leaf area (Mishra, 1997). Data is presented as the mean of 15 leaf observations within each line or control plants for length, density and stomatal index.

Chromosome count

Mitotic chromosomes were prepared from young root meristems, following the method described by Las Peñas, Bernardello & Kiesling (2008), with minor modifications. In vitro root tips (∼0.5–1 cm) were excised from putative polyploid lines and diploid controls between 7 and 8 o’clock in the morning and pretreated with 0.002 mM 8-hydroxyquinoline (8-HQ) for 24 h at 4 °C in the dark and rinsed with distilled water for 5 min to remove 8-HQ. The root tips were then fixed in Farmer’s solution (ethyl alcohol: acetic acid, 3:1 v/v) for 24 h at room temperature and rinsed with distilled water. Subsequently, samples were hydrolyzed in 1N HCl at 60 °C for 10 min and stained with Schiff’s reagent for 1 h in the dark. Finally, each hydrolyzed root was crushed in a drop of 45% (w/v) acetocarmine and 45% (w/v) acetic acid, and the number of chromosomes in mitotic cells was determined using a light microscope at 100x magnification (DM500®; Leica Microsystems, Wetzlar, Germany). A total of 10 representative photomicrographs were analyzed from three root tips from each line, including the control.

Flow cytometry

Fresh apical leaves from the selected in vitro lines, including the control, were chopped in 1.5 mL Galbraith’s modified buffer (45 mM MgCl2, 30 mM sodium citrate, 20 mM 4-morpholine propane sulfonate (MOPS) and 0.5% (v/v) TritonX-100, pH 7.0) (Galbraith et al., 1983). After filtration through a 30 μm nylon mesh, crude nuclear samples were stained with 10 mg of propidium iodide. Nuclear DNA content was determined, using the method described by Arumuganathan & Earle (1991), which employs an Attune® Acoustic Focusing Flow Cytometer blue/violet (Applied Biosystems, Foster City, CA, United States of America). Fresh leaves from Solanum lycopersicum (2C = 1.96 pg DNA) were used as an internal standard (Doležel, Greilhuber & Suda, 2007) and more than 5,000 nuclei per sample were analyzed. Three independent replicates were performed for each analysis.

Floral characteristics

To study the effect of polyploidy on some floral characteristics, the size of the flower, as well as the color were assessed in ten tetraploid lines and one control line. Three plants were chosen per each line and within each plant two inflorescences at peak bloom were randomly. Eighteen individual, fully opened flowers from the top, middle, and bottom of the inflorescences were selected for flower length and maximum calyx length, using a digital vernier caliper. Color coordinates were performed using a sprectrophotometer (X-Rite SP64; X-Rite, Grand Rapids, MI, USA) (McGuire, 1992). The X-Rite SP64 was positioned with minimal pressure, perpendicular to the lower lip of each flower and the data were reported in the L* (luminosity 0 = black, 100 = white), C* (chromaticity, saturation level of h) and h (tone angle: 0° = red, 90° = yellow, 180° = green, 270° = blue, 300° magenta) colorimetric system according to Commission Internationale de l’éclairage (CIE) (Schanda, 2007). For accuracy comparison Royal Horticultural Society (RHS) Colour Charts (Royal Horticultural Society, 2015) were used to compare with the X-Rite SP64 sensor readings. Data reported represent the averages for three measurements per flower.

Leaf area and chlorophyll content

Ten tetraploid lines were evaluated under greenhouse conditions to assess the effect of polyploidy on leaf area and chlorophyll content. Leaf area was determined with a leaf area meter (LI-3100C AREA METER; LI-COR® Bio Sciences Instrument, Lincoln, NE, USA) and chlorophyll content (Chl a, b and total) using a portable chlorophyllmeter ClorofiLOG (model CFL 1030; ClorofiLOG, Falker, Brazil). For each line and control (diploid), five plants in vegetative phase were selected at random, and triplicate measurements were performed on two fully developed leaves, taken from the middle section of the shoots (n = 11 lines including the control × five plants × two leaves) (Xavier et al., 2021).

Quantification of tilianin by HPLC

The quantification of tilianin from tetraploid and diploid lines was based on the method established by Hernández-Abreu et al. (2009). For each line in vegetative phase of plant development, 10 g of finely ground dried plant material (leaves and stems) was subjected to continuous maceration (1:10 w/v) with hexane (C6H14), dichloromethane (CH2Cl2), and methanol (CH30H) three times for 72 h at room temperature. The tilianin content was determined in the methanolic extract using Waters HPLC equipment with a photodiode array detector, Zorbax C18 SB-CN (4.6 mm × 250 mm, 5-μm particle size; Agilent®), and data analysis was performed using Empower 2002 software. The mobile phase consisted of methanol-water at a ratio of 61:39 (v/v) at a flow rate of 0.7 ml/min and a wavelength of 260 nm. The quantification of tilianin was defined according to the corresponding calibration curve at concentrations of 103.258, 34.264, 27.693, 20.173, 13.027, and 6.442 μg/mL, using highly purified tilianin as reference. All samples were assayed in triplicate.

Statistical analysis

The effect of different concentration of colchicine (0.0, 0.1, 0.3, 0.5%) and exposure time (6, 12, 24 h) on seedling survival rate was analyzed by applying a multiple linear regression (Rossi, 2022). To determine possible phenotypic differences between ten polyploid lines (genotypes) and diploid control, we performed one-way analysis of variance (ANOVA) to define stomata length, stomata density, stomata index, leaf area, chlorophyll a, chlorophyll b, total chlorophyll and tilianin content. Differences were analyzed by applying the Tukey multiple comparisons test (p < 0.05). Principal component analysis (PCA) was carried out to determine potential relationships between genome size and phenotypic traits, previously mentioned. All analyses were performed in R version 4.2.3 (R Core Team, 2023) using ade4 (PCA, Dray & Dufour, 2007) and ggplot2 (Wickham, 2016) libraries, which were used to generate graphs.

Results

Identification of tetraploids and ploidy stability

After 11 days of in vitro culture, 29.8% of the A. mexicana seeds germinated. Seedlings at the stage of cotyledon emergence were selected for the in vitro polyploid induction using colchicine. The first visible effect of colchicine was reflected in the browning and delayed growth rate of treated seedlings. Significant differences between treatments were observed after 2 weeks of culture, and multiple linear regression analysis (y = 1.278-colchicine (2.279)-time exposure (0.005)) revealed a strong negative correlation between seedling survival and colchicine dosage (p = 0.003), but no significant correlation was found with exposure time. The second observed effect of colchicine was the inhibition of two processes: (1) shoot formation from axillary buds in seedlings, and (2) in vitro rooting of excised shoots to produce plantlets. The number of shoots per seedling decreased with increasing colchicine dose, resulting in a shoot count of 0.40 ± 0.329 for colchicine-treated seedlings and 1.79 ± 0.318 for untreated (diploid control) seedlings after 4 weeks in culture. Overall, higher colchicine doses corresponded to lower survival rates and fewer plantlets produced (putative polyploid lines) (Table 1).

Table 1 Survival rate and polyploidy level of Agastache mexicana lines obtained after polyploidization treatments.

Treatment	Survival rate (%)	Number of regenerated plantlets (Lines)	Ploidy level determined by flow cytometry	
Colchicine %	Time (h)	No. of diploid lines	No. of tetraploid lines (%)*	No. of mixoploid lines (%)*	
0	6	100	19	19	0	0	
0	12	100	19	19	0	0	
0	24	100	21	21	0	0	
0.1	6	58.33	12	1	11 (91.66)	0	
0.1	12	50.00	7	2	4 (57.14)	1 (14.28)	
0.1	24	8.33	7	1	6 (85.71)	0	
0.3	6	8.33	0	0	0	0	
0.3	12	0	0	0	0	0	
0.3	24	16.66	0	0	0	0	
0.5	6	8.33	3	1	2 (66.66)	0	
0.5	12	8.33	0	0	0	0	
0.5	24	8.33	0	0	0	0	
Note:

* Efficiency of polyploidization. Twelve seedlings were used per treatment.

In this study, 29 putative polyploid lines of A. mexicana were obtained from colchicine treatments. Generally, after 8 weeks of in vitro culture, the putative polyploid plantlets showed markedly different morphological characteristics from the diploid controls, manifesting smaller or larger leaves, often with a rolled structure, and ranging in color from purple to dark green color. Significant differences were observed between the stomatal characteristics of the in vitro diploid control and tetraploid lines. As shown in Fig. 2, the leaves from tetraploid lines exhibited large stomata, with significantly higher stomatal lengths than leaves from diploid control (F5,24 = 4.613, p < 0.01; 4x = 37.94 ± 7.81 μm; 2x = 31.46 ± 4.00 μm), as well as, a lower stomatal density (F5,24 = 21.82, p < 0.001; 4x = 117.5 ± 60.26 PVF; 2x = 161.24 ± 19.43 PVF) and stomatal index (F5,24 = 18.47, p < 0.001; 4x = 10.25 ± 2.22%; 2x = 16.23 ± 1.85%).

Figure 2 Comparison of the leaf and stomata characteristics between diploid and induced tetraploid plants.

Diploid (A and C) and tetraploid plant (B and D). Photographs correspond to representative samples. Five leaves from each plant were used for stomatal observations.

Studies of mitotic cells in actively growing root tips clearly indicated that the polyploidy of the samples was due to chromosome doubling in diploid seedlings induced by colchicine treatments. Figure 3 shows that the number of chromosomes in the polyploid plantlets is 2n = 4x = 36, whereas the number of chromosomes in the diploid plantlets (controls) is 2n = 2x = 18. Of the 29 lines evaluated; 23 (79.3%) were tetraploids, 5 (17.3%) diploids, and 1 (3.4%) mixoploid, containing a mixture of diploid and tetraploid cells. In order to confirm the ploidy level of plantlets using flow cytometry, as there have been no reports to date that estimate genome size of A. mexicana, it was necessary to determine the genomic content, by comparing the mean position of the G0/G1 peak of the internal standard of S. lycopersicum with the mean position of the peak of the diploid sample of A. mexicana. Eight independent measurements were performed to conclude that diploid control plants (2n = 18) have an average genomic content of 1.433 ± 0.025 pg. Flow cytometry DNA histograms of the polyploid lines revealed that the peak position was twice that of the genome DNA of diploid (Fig. 4). It was thus concluded that the DNA content of the tetraploid cells (2.660 ± 0.236 pg) equals twice that diploid cells. The three groups of ploidy levels: diploid, tetraploid, and mixoploid plantlets based on flow cytometry, fully concurred with the results obtained from counting chromosomes. The highest percentage of tetraploid production efficiency (91.66%) was achieved with 0.1% colchicine for 6 h, without generating any mixoploids (Table 1). Chromosome counting and flow cytometry analysis indicated that DNA ploidy levels remain stable in all tetraploid lines after 15 months of in vitro culture.

Figure 3 Chromosome counts of A. mexicana.

(A) Tetraploid cell with 36 chromosomes and (B) diploid cell tetraploid cell of A. mexicana with 18 chromosomes. Ten representative photomicrographs from three root tips per line were analyzed.

Figure 4 Flow cytometry analysis of A. mexicana using Solanum lycopersicum as the internal standard (IS).

(A) Diploid, (B) tetraploid, and (C) induced mixoploid A. mexicana plant. Three independent replicates were performed for each analysis.

During in vitro culture, the mixoploid line and several tetraploid lines showed symptoms of hyperhydricity, low growth capacity, and reduced root development (Fig. 5). Consequently, during the acclimatization process, the mixoploid plants did not survive the transfer from in vitro conditions to greenhouse conditions. In contrast, ten tetraploid lines exhibited accelerated growth and development in comparison to the diploid control. These lines were selected to evaluate the effect of ploidy on leaf and flower traits, as well as and on tilianin content in A. mexicana.

Figure 5 Comparison of morphological characteristics of in vitro plants of A. mexicana.

Diploid (A), tetraploid (B), and mixoploid (C).

Flower and leaf characteristics of A. mexicana grown under greenhouse condition

Inflorescence emergence began earlier in the tetraploid plants; after only 3 months in the greenhouse, compared to 4 months for diploid control. All tetraploid lines, except for line Amx2, exhibited significantly increased flower and calyx lengths compared to the diploid control (Table 2; Fig. 6). In tetraploid lines, the mean lengths of flowers and calyxes were 30.659 ± 2.401 mm and 11.732 ± 0.801 mm, respectively, so significantly greater than those in diploid lines (25.252 ± 2.947 mm and 11.271 ± 2.334 mm, respectively; p < 0.0001). Also, there were significant differences in the color components of flowers of tetraploids and diploid control (Table 2). However, in both tetraploid lines and diploids, the value of L* tends to be neutral, the color tends to be magenta and slightly more opaque (C* = less than 30) in the tetraploid lines, compared to the diploid control, which tends to be violet. This resembles the visual evaluation provided by the RHS, which identified the color of the diploid flowers as deep strong reddish purple C (NN78) and the tetraploid flowers as light reddish purple D (NN78).

Table 2 Petal color and flower size of 10 tetraploid lines and a diploid control grown in the greenhouse.

Line	Flower length (mm)	Calyx length (mm)	Color parameters	
L*	C*	h	RHS Coulor chart	
Control	25.252 ± 2.947d	11.271 ± 2.334cd	45.735 ± 2.831d	32.336 ± 1.909a	319.822 ± 1.481b	Strong reddish purple C(NN78)	
Amx 1	29.967 ± 1.759bc	11.294 ± 0.596cd	50.962 ± 2.356ab	26.864 ± 2.154d	321.572 ± 2.616ab	Light reddish purple D(NN78)	
Amx 2	25.216 ± 3.599d	10.624 ± 0.925d	50.224 ± 4.043ab	27.472 ± 4.371d	323.039 ± 1.806a	Light reddish purple D(NN78)	
Amx 3	30.842 ± 2.588bc	11.092 ± 0.716cd	46.126 ± 1.661d	31.033 ±1.745ab	323.238 ± 1.291a	Light reddish purple D(NN78)	
Amx 4	30.555 ± 3.873bc	11.486 ± 0.819cd	49.639 ± 4.141bc	30.579 ± 2.877abc	322. 366 ± 2.870ab	Light reddish purple D(NN78)	
Amx 5	31.622 ± 1.987ab	12.609 ± 0.484ab	51.238 ± 2.573ab	28.915 ± 1.765bcd	323.933 ± 1.258a	Light reddish purple D(NN78)	
Amx 6	31.006 ± 2.631bc	13.347 ± 0.843a	50.878 ± 3.685ab	27.984 ± 2.133dc	322.306 ± 2.870ab	Light reddish purple D(NN78)	
Amx 7	32.466 ± 0.832ab	12.026 ± 1.048bc	51.124 ± 3.532ab	27.047 ± 3.421d	322.85 ± 1.657a	Light reddish purple D(NN78)	
Amx 8	34.094 ± 1.598a	11.894 ± 0.449bc	52.871 ± 1.451a	26.573 ± 2.291d	324.094 ± 6.688a	Light reddish purple D(NN78)	
Amx 9	32.076 ± 3.609ab	11.842 ± 0.697bc	51.519 ± 2.175ab	28.642 ± 1.4bcd	321.483 ± 1.637ab	Light reddish purple D(NN78)	
Amx 10	28.751 ± 2.134c	11.113 ± 0.553cd	46.721 ± 1.559dc	31.182 ± 1.321ab	324.006 ± 0.803a	Light reddish purple D(NN78)	
ANOVA	F10,187 = 19.89, p < 0.0001	F10,187 = 10.86, p < 0.0001	F10,187 = 12.59, p < 0.0001	F10,187 = 11.89, p < 0.0001	F10,187 = 3.993, p < 0.0001		
Note:

Different letters indicate significant statistical differences between samples, defined by one-way ANOVA followed by Tukey’s test for multiple comparisons at p < 0.05, with results referring to the mean of three observations ± SD. L* (Lightness), C* (Chromaticity), h (hue angle), RHS, Royal Horticultural Society.

Figure 6 Comparison of flower characteristics between diploid and induced tetraploid plants.

Diploid (A and C) and tetraploid plant (B and D). Photographs correspond to representative samples. Eighteen flowers from each plant type were analyzed for size and color.

Differences in leaf area were observed between the diploid and the tetraploid lines Amx3, Amx7, and Amx10, with the tetraploids manifesting a higher chlorophyll content and smaller leaf area (Fig. 2, Table 3). Tetraploid plants presented a dark green leaf color.

Table 3 Leaf area and chlorophyll content in mature leaves from 10 tetraploid lines and a diploid control grown in the greenhouse.

Line	Leaf area
(cm2)	Total chlorophyll	Chlorophyll a	Chlorophyll b	
Control	8.561 ± 1.433ab	363.7 ± 35.78bc	278.6 ± 22.657bc	85.1 ± 15.807b	
Amx 1	8.596 ± 3.999ab	363.5 ± 21.072bc	271.8 ± 26.317bc	91.7 ± 12.074ab	
Amx 2	9.32 ± 3.377a	414.1 ± 24.269ab	296.1 ± 25.339abc	118.0 ± 13.241a	
Amx 3	4.152 ± 1.413d	390.8 ± 87.152abc	297.9 ± 47.082abc	92.9 ± 43.193ab	
Amx 4	8.205 ± 1.548abd	345.6 ± 29.349c	258.6 ± 22.061c	87.0 ± 9.428b	
Amx 5	8.282 ± 3.465abd	391.9 ± 38.155abc	295.9 ± 30.555abc	95.8 ± 11.849ab	
Amx 6	5.712 ± 1.238bcd	421.8 ± 39.202ab	319.2 ± 35.266ab	102.6 ± 9.901ab	
Amx 7	4.570 ± 1.307d	443.5 ± 73.971a	333.4 ± 70.298a	110.1 ± 15.058ab	
Amx 8	10.833 ± 3.066a	383.7 ± 21.638abc	294.1 ± 11.551abc	89.6 ± 11.568ab	
Amx 9	8.675 ± 2.921ab	385.6 ± 33.731abc	296.4 ± 21.593abc	89.2 ± 12.951ab	
Amx 10	4.93 ± 1.216cd	411.5 ± 49.365abc	314.6 ± 23.324ab	96.9 ± 34.027ab	
ANOVA	F10,115 = 8.793,
p < 0.0001	F10,99 = 3.915,
p < 0.001	F10,99 = 3.929,
p < 0.001	F10,99 = 2.575,
p < 0.01	
Note:

Different letters indicate significant statistical differences between samples, defined by one-way ANOVA followed by Tukey’s test for multiple comparisons at p < 0.05. All samples were processed with three independent measurements.

Tilianin content

Statistically significant differences in tilianin content were found between the tetraploid lines and diploid control in methanolic extract (F10, 22 = 23,806, p < 0.0001). The diploid control produced an average of 6.388 ± 0.005 mg of tilianin per gram of dry weight (mg/g DW). Out of the tetraploid lines, the greatest tilianin accumulation was recorded in Amx3 (32.964 ± 0.004 mg/g DW) and Amx7 (32.392 ± 0.110 mg/g DW). Compared to the control (as shown in Fig. 7), Amx8 (3.195 ± 0.005 mg/g DW) exhibited less accumulation of tilianin.

Figure 7 Total tilianin content in control and induced polyploidy (Amx 1–10) lines of A. mexicana.

Different letters on the vertical bars differ significantly (Tukey HSD Test, p < 0.05). The assay was conducted in triplicate for each simple.

Principal component analysis

Principal component analysis (PCA) was performed to investigate how the patterns of variance of morphological traits and tilianin content correlated with ploidy level. This was also used to determine their possible contribution to the content of tilianin and to identify particular traits that distinguish diploid from tetraploid plants. Two principal components, PC1 and PC2, were determined, based on their degree of contribution. The first two components accounted for 57.05% of total variance, with 33.88% relating to PC1 and 23.17% relating to PC2 (Fig. 6). PC1 correlated positively with C* (0.711) and negatively with calyx length (−0.805), DNA (−0.796), flower length (−0.760) and h (−0.498). PC2 showed a positive correlation with tilianin (0.714), and negative correlation with L (−0.748) and leaf area (−0.589). The control group is separated from the tetraploid lines in Fig. 8B.

Figure 8 Principal component analysis of tetraploid and diploid control of A. mexicana.

(A) Separation of the variables as a function of tetraploid and control lines. (B) PC1 vs. PC2, with 57.05% variance. DNA; tilianin; Chlo T, total chlorophyll; LA, leaf area; FL, flower length; CL, calyx length; L, lightness; C*, chromaticity; h, hue angle.

Discussion

Antimitotic agents such as colchicine have been used in medicinal plant breeding for artificial in vitro polyploidy induction and the development of varieties with improved agronomic traits, enhanced abiotic stress tolerance, and high levels of bioactive secondary metabolites. However, the phenotypic and genetic changes in plants that occur due to the artificial chromosome doubling are often unpredictable and can differ significantly between species (Niazian & Nalousi, 2020; Tavan et al., 2022). In this study, the feasibility and efficacy of colchicine for polyploidization of A. mexicana and the generation of polyploid lines with high tilianin production were investigated. For this purpose, we follow the techniques previously applied to other plant species, such as Platanus aceriolia (Liu, Li & Bao, 2007), Arabidopsis thaliana (Yu et al., 2009), and Agastache foeniculum (Talebi et al., 2017). In vitro-cultured young seedlings of A. mexicana were directly treated with one of four colchicine concentrations (0.0, 0.1, 0.3, and 0.5%, w/v) for 6, 12 and, 24 h. The results from the present study revealed that the survival of seedlings and the efficiency of tetraploid lines recovery were affected by the concentration of this antimitotic agent but not by the duration of exposure (Table 1). Some studies suggest that colchicine has a low affinity for plant tubulin, requiring relatively high concentrations to disrupt microtubule formation and consequently promote polyploidization (Hailu et al., 2021). However, high concentrations of colchicine have been shown to alter gene expression, including genes related to the phenylpropanoid biosynthetic pathways and plant hormone signaling. This alteration may contribute to explant mortality during chromosome doubling and low plantlet development ability caused by the death of meristematic cells (Temel & Gozukirmizi, 2015; Zhou et al., 2017). It is therefore paramount to carefully select a suitable colchicine concentration to achieve a high polyploid induction rate. Indeed, some research suggests that factors, such as plant genotype, ecotypes, type and age of explants are also important for improving artificial polyploidy induction efficiency (Salma, Kundu & Mandal, 2017; Niazian & Nalousi, 2020). In this study, the most efficient treatment for tetraploid induction of A. mexicana was observed to be 0.1% colchicine for 6 h, with a 91.66% explant survival rate and 58.33% tetraploidy induction. These results concur with previous studies showing that colchicine concentrations used for the induction of polyploid plants usually range from 0.005% to 1.0% (w/v), and the duration of treatment takes from hours to weeks, depending on application methods (Eng & Ho, 2019; Ahmadi & Ebrahimzadeh, 2020).

Microscopy examination of meristematic cells showed that the diploid plants of A. mexicana have a chromosome number of 2n = 18, whereas the tetraploid lines plants have 4n = 36 (Fig. 3). This result concurred with the previous report of a haploid number n = 9 for A. mexicana (Sanders, 1987). Polyploidy was confirmed using flow cytometry, a method which proved to be fast and accurate for estimating the increase in DNA content in A. mexicana (Fig. 4). The DNA content estimated for diploid plants (2C = 1.433 ± 0.025 pg) and tetraploid plants (2.660 ± 0.236 pg) of A. mexicana was similar to that reported for A. foeniculum. As far as we know, among the Agastache genus, A. foeniculum is the only species for which the genome sizes have been reported; with means of 1.06 ± 0.02 pg for diploid 2C value and 2.15 ± 0.001 pg for tetraploid plants (Talebi et al., 2016).

Studies on several species suggest that the extra set of chromosomes in polyploid plants will often, though not always, lead to increased biomass or content of bioactive secondary metabolites, by means of changes in gene transcription, epigenetic modifications, and morphological and physiological alterations (Osborn et al., 2003; Lavania, 2013; Iannicelli et al., 2020); therefore, the phenotypic variations generated by the polyploidization may help distinguish diploid from polyploid plants. In this study, under in vitro conditions, the results indicated that an initial screening on the basis of stomata size might be effective for identifying putative polyploids. According to Beaulieu et al. (2008), there is a positive correlation between genome size and guard cell length and a negative correlation between stomatal size and stomatal density. This concurs with the results presented here, indicating that an increase in ploidy level increases stomatal length and decreases stomatal density and stomatal index in A. mexicana autotetraploid plants (Fig. 2). Similar to previous studies reported in tetraploid plants of Hibiscus syriacus (Lattier, Chen & Contreras, 2019) and A. foeniculum (Talebi et al., 2017).

The diploid control, tetraploid, and mixoploid lines showed slight symptoms of hyperhydricity, a common problem for in vitro cultures (Gao et al., 2017), which may have affected rooting and greenhouse acclimatization (Fig. 5). Hyperhydricity also resulted in delayed development of the only mixoploid line obtained. This line eventually ceased to grow after a brief period of time and did not survive the transfer from in vitro conditions to greenhouse. Consequently, only the stable tetraploid plants, confirmed by flow cytometry after 15 months of successive in vitro subcultures were selected for transfer to the greenhouse to determine leaf area, chlorophyll content, flower characteristics, and tilianin content.

The PCA analysis indicates a positive correlation between flower size and DNA content of tetraploid plants under greenhouse conditions (Fig. 8). A. mexicana tetraploid plants exhibited larger flowers than their diploid counterparts (Fig. 6). Flowers from diploids plants were smaller and tended to be purple compared to tetraploid plants, which tended to be magenta (Table 2). Previous findings reported flower color modifications among tetraploid plants of Rosa centifolia, Impatiens walleriana and probably Gladiolus grandiflorus, resulting from alterations in the biosynthesis pathway of pigments (e.g., anthocyanin) and increased size of cells due to chromosome doubling effect (Osborn et al., 2003; Manzoor et al., 2018; Ghanbari et al., 2019). It is generally accepted that the cells in polyploid plants tend to be larger than those in their corresponding diploid plants, resulting in thicker and bigger leaves, as well as larger flowers and fruits (Hu et al., 2021). Increase in flower size has been observed in the autopolyploid plants of several species, including Crocosmia aurea (Hannweg, Sippel & Bertling, 2013), Gerbera jamesonii Bolus cv. Sciella (Gantait et al., 2011), Hemerocallis x hybrida ‘Blink of an Eye’ (Podwyszyńska et al., 2015) and, Vicia villosa (Tulay & Unal, 2010). However, in most tetraploid lines of A. mexicana with large flowers, there was a tendency for the leaf area to decrease, as shown by PCA analysis (Fig. 8). Previous studies have noted that among certain species, including Rhododendron furtunei (Mo et al., 2020), Escallonia rosea (Denaeghel et al., 2018), Gladiolus grandiflorus (Manzoor et al., 2018), and Arabidopsis thaliana, the leaf area of tetraploid and octoploid variants decreases as their ploidy level increases. According to Iannicelli et al. (2020), the polyploid cells may expand to maintain the balance between cytoplasmic and nuclear volume, which is necessary for efficient cellular machinery function, due to an increased DNA content in the nucleus. This results in a delay in the growth of tissues and organs due to higher energy expenditure and a decrease in the surface area to volume ratio of polyploid cells, thereby reducing growth and cell division (Tsukaya, 2008; Aqafarini et al., 2019).

On the other hand, a large number of studies have demonstrated that changes in plant ploidy affect the gene transcripts, protein synthesis, and photosynthetic elements, which often result in a beneficial impact on secondary metabolite biosynthesis (Lavania et al., 2012; Gantait & Mukherjee, 2021). Interestingly, PCA analysis revealed that DNA content did not correlate with the total chlorophyll and tilianin content in A. mexicana. However, this statistical analysis demonstrated a significant negative correlation between the accumulation of tilianin and leaf area (Fig. 8). The tetraploid lines Amx3 and Amx7, which have the smallest leaf area, showed higher tilianin content than diploid plants. In contrast, plants with larger leaves had lower tilianin accumulation than the diploid controls (Table 3, Fig. 7). Since previous research has shown that the exogenous application of benzylaminopurine increases tilianin content (Carmona-Castro et al., 2019) and results in plants with significantly smaller leaves compared to A. mexicana plants cultivated on substrates without cytokinins (Copetta et al., 2023), we hypothesized that the lines Amx3 and Amx7 may have a high cytokinins content, facilitating greater accumulation of this flavonoid. Similarly, in earlier studies have demonstrated that cytokinins can affect secondary metabolites accumulation. For instances, cytokinins have been reported to increase flavonoid production in Scutellaria altissima shoot cultures (Grzegorczyk-Karolak, Kuźma & Wysokińska, 2017), and root cultures of Morinda citrifolia (Baque, Hahn & Paek, 2010). This effect possibly results from the interaction of plant growth regulators with biosynthetic enzymes in secondary metabolic pathways, such as phenylalanine ammonia-lyase, chalcone synthase, and geranyltransferase (Jamwal, Bhattacharya & Puri, 2018). Likewise, Powell & Doyle (2015) reported that biomass changes in biosynthetic tissues of polyploid plants may either increase or decrease flavonoid production. Both qualitative and quantitative variations in flavonoids levels have been detected in various autotetraploid plants, including Chamomilla recutita, Isatis indigotica, and Polemonium caeruleum (Repák, 2000; Zhang et al., 2021; Samatadze et al., 2022). However, the molecular mechanisms that explain the association between polyploidy and flavonoid metabolism remain undefined. Polyploidy involves more than just chromosome doubling and typically results in changes to genome structure and gene expression (Xing et al., 2024; Doyle & Coate, 2019). Information about the morphological traits of polyploid plants of Agastache genus and their effect on phenolic and flavonoid compounds remains limited. In the future, the study of genetic variation in the expression of gene encoding enzymes involved in the tilianin pathways will shed light on the regulatory mechanism by which the ploidy level affects the flavonoids content in A. mexicana.

Conclusions

In conclusion, the results indicate that artificial induction of autopolyploidy in Agastache mexicana was successful in obtaining different tetraploid genotypes with phenotypic characteristics distinct from those of the diploid plants. The changes in morphology and bioactive compound content of the polyploid plants could represent an agronomic advantage for their cultivation. Furthermore, it is important to note that tetraploid lines of A. mexicana (Amx3 and Amx7) exhibited higher tilianin content compared to diploid plants. These findings highlight the significance of plant polyploidy in enhancing secondary metabolite production. Understanding the relationship between leaf characteristics, cytokinin content, and flavonoid accumulation can provide valuable insights for optimizing tilianin production in tetraploid A. mexicana. Further investigation into the regulation of genetic and biochemical pathways involved in tilianin biosynthesis will be essential for advancing our knowledge in this area.

Supplemental Information

Supplemental Information 1 Total tilianin contents of the diploid control and tetraploid lines.

Each sample was analyzed in triplicate. (mg/g dry weight of the plant)

Supplemental Information 2 Input data for PCA analysis.

DNA content; tilianin content; Chlo T, total chlorophyll; LA, leaf area; FL, flower length; CL, calyx length; L, lightness; C*, chromaticity; h, hue angle.

Supplemental Information 3 Input data for ANOVA analysis to characterize flowers in tetraploid lines and control.

L, lightness; C, chromaticity; h, hue angle.

Supplemental Information 4 Input data for ANOVA analysis to characterize leaves in tetraploid lines and control.

The authors are thankful to Ariadna Zenil Rodríguez from Centro de Investigación en Biotecnología for her help with the HPLC Waters, as well as to Doctorado en Ciencias Naturales de la Universidad Autónoma del Estado de Morelos. The authors appreciate the collaboration of Laboratorio Nacional de Investigación y Servicio Agroalimentario y Forestal, Universidad Autónoma de Chapingo and the technical support of Juan Carlos Cervantes Gutiérrez of Flow Cytometry Area.

Additional Information and Declarations

Competing Interests

Author Contributions

Data Availability

The authors declare that they have no competing interests.

Angélica Martínez-Aguilar conceived and designed the experiments, performed the experiments, analyzed the data, prepared figures and/or tables, authored or reviewed drafts of the article, and approved the final draft.

Evert Villanueva Sánchez conceived and designed the experiments, performed the experiments, analyzed the data, prepared figures and/or tables, authored or reviewed drafts of the article, and approved the final draft.

Susana Valencia-Díaz conceived and designed the experiments, analyzed the data, prepared figures and/or tables, authored or reviewed drafts of the article, and approved the final draft.

Samuel E. Estrada-Soto conceived and designed the experiments, analyzed the data, prepared figures and/or tables, authored or reviewed drafts of the article, and approved the final draft.

Selene Napsucialy-Mendivil conceived and designed the experiments, performed the experiments, analyzed the data, prepared figures and/or tables, authored or reviewed drafts of the article, and approved the final draft.

Rodrigo Barba-Gonzalez conceived and designed the experiments, analyzed the data, authored or reviewed drafts of the article, and approved the final draft.

Iran Alia-Tejacal conceived and designed the experiments, analyzed the data, authored or reviewed drafts of the article, and approved the final draft.

José de Jesús Arellano-García conceived and designed the experiments, analyzed the data, authored or reviewed drafts of the article, and approved the final draft.

Oscar Gabriel Villegas Torres analyzed the data, authored or reviewed drafts of the article, and approved the final draft.

Karla Catalina Cruz Torres performed the experiments, analyzed the data, authored or reviewed drafts of the article, and approved the final draft.

Irene Perea-Arango conceived and designed the experiments, performed the experiments, analyzed the data, prepared figures and/or tables, authored or reviewed drafts of the article, and approved the final draft.

The following information was supplied regarding data availability:

The raw data used for the statistical analyses are available in the Supplemental Files.

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
