# Peer review of "Tilianin content and morphological characterization of colchicine-induced autotetraploids in Agastache mexicana"

_PeerJ, doi:10.7717/peerj.18545_

## Round 0.1 · original submission · Major Revisions

Thank you authors for your patience as your work was evaluated by scholarly reviewers. Please, you can see they found your work very promising, and have identified areas of concern. Kindly address them to your best detail, and give strong attention to the experimental design, and the discussion. Ensure to provide detailed responses to reviewers' comments, and the same detail in the revised manuscript. Look forward to the improved revised work.

·

Basic reporting

Acceptable.

Experimental design

Appropriate and minor corrections are required. Refer to the annotated pdf file.

Validity of the findings

Appropriate. Some of the statements are unclear, and minor corrections are required. Refer to the annotated pdf file.

Additional comments

This study aims to produce autotetraploid lines with increased tilianin production by using colchicine. The results are interesting as only one mixoploid line (14.28%) was observed during the experiment compared to other published studies. It is recommended to further investigate the morphological characteristics and performance of this mixoploid, and if the results are available, it would be beneficial to include them in the manuscript besides the results of diploid and tetraploid.

Reviewer 2 ·

Basic reporting

- In the introduction, I suggest including relevant aspects of in vitro reproduction of lemon balm plants, in addition to the production of tilianin, to justify the reason for using this technique instead of germinating the seeds in some substrate.

- In Table 2, check if it refers to flowers or inflorescences, and verify if the units are in cm or mm.

Experimental design

- In Mexico, there are several plants that are considered lemon balm "toronjil", and the same species reported in the article even has two subspecies. It is necessary to indicate which one the authors are referring to. From the color of the inflorescences shown in one of their tables, it is inferred that it is the Mexicana subspecies, but it needs to be specified.

- In line 120, reference is made to seeds for carrying out in vitro reproduction. However, they do not detail the origin of the seeds or the plants from which those seeds came. Were they from a previous harvest? Given the constant confusion surrounding the species and subspecies of "lemon balm", it is recommended to include the herbarium specimen number detailing the species and subspecies in question. It would also be interesting to include the germination percentage in the reported medium, as this particular species is not known for its seed reproduction.

-In line 124, specify that the temperature used was only that of the medium, without involving the plant, otherwise it would be harming the plant.

- In line 148, they mention that the plants were cultivated in a greenhouse. I suggest including details about the phenological stage of obtaining the plant material for the phytochemical tests.

- In line 220, specify whether the aerial part of the plant includes leaves and inflorescences.

- In line 254, check if the correct term is "seedlings" or if it refers to the generation of new shoots.
- In line 293, reference is made to greenhouse cultivation, but the characteristics of the cultivation are not detailed in the materials and methods, such as whether it was in soil or hydroponics, etc.

Validity of the findings

The proposal is interesting and shows hard work. There are details that can be improved. I suggest to the authors to thoroughly review the aspects I suggest.

Additional comments

Lemon balm is a species that has been relatively underexplored in terms of details regarding its in vitro reproduction and greenhouse growth. I invite the authors to include more details on this matter, which would clarify the origin of the plant material used in the research and provide greater solidity for the generation of knowledge about this species, particularly the Mexican subspecies.

Reviewer 3 ·

Basic reporting

This manuscript reports a study conducted for chemical induction of autotetraploid plants from diploid Agastache mecicana, characterization of selected tetraploid plants including the concentration of tilianin, a bioactive compound with pharmaceutical value. The authors are commended for their success in production of some tetraploid Agastache mecicana that contained much higher concentration of tilianin than the diploid parental plants.

Introduction section appears to be appropriate since rationales for conducting this study were elucidated. Please consider breaking it to three paragraphs: Agastache mecicana plants and medicinal value, ploidy level manipulation, and objectives.

Major concerns are in the Materials and Methods section which will be mentioned in 2. Experimental design.

Result section:
(1) Figure 1 legend: Mention five leaves were used for stomatal observations.
(2) Figure 2 legend: Mention 10 representative photomicrographs were analyzed from three root tips per line.
(3) Figure 3 legend: Mention three replicate.
(4) Figure 4 legend: Mention three replicate.
(5) Table 3. 10 tetraploid lines were used, any replication per line?

Discussion section:
(1) Did you compare leaf thickness among lines? Amx3 and 7 might had rather thicker leaves than the others.
(2) Please provide your speculation as to why smaller leaved tetraploids (Amx3 and 7) contained higher tilianin?

More information is needed for the Conclusion section.

References
Additional references may need for speculation of the final question under Discussion.

Experimental design

(1) What was experimental design and replication?
(2) It is unclear how those polyploid plants were produced. “the newly regenerated shoots” is not correct term. Regeneration is the production of plants from non-meristematic tissue, which is through shoot organogenesis or somatic embryogenesis. Since the culture medium had no growth regulators, I assume it was shoot culture, meaning shoots came from existing meristem, for example, at node. The authors should make this clear.
(3) References should be provided for analysis of chlorophyll contents.
(4) It is important to first mention experimental design. Without design, how do you conduct statistical analysis.

Validity of the findings

The results support the claim, but the authors need to make appropriate revision based on the concern mentioned above.

---

## Round 0.2 · Minor Revisions

Thank you authors for your patience. Feedback from reviewer is positive toward the publication of your revised work. To further elevate the quality of this work, please, kindly consider addressing the following:

a)The introduction needs additional information. It will be worthwhile to include the following: i) The importance of polyploid induction, and its implication in plants, citing instances; ii) the various ways polyploidization events on chlorophyll contents are measured, and why they are important. Please, make sure to have this as a new paragraph just before current line 121.
b)The materials and methods need additional information. Please, this section should start with a new subsection captioned 'Schematic overview of the experimental program", which should comprise 5-6 sentences and must be supported by a flow diagram that shows how this study was conducted, from collection of plant parts, all the way to analytical methods. The first sentence should introduce the schematic diagram, the second and third should describe the key stages, fourth and fifth should connect the diagram with the specific aim of this work, and the assurance that all methods were in adherence to standard laboratory procedures
c) Given that results and discussion are separate sections, please authors should go through the discussion, and make sure that all the tables and figures presented in the results are captured in the discussion. All must be captured, and please in the discussion, use (Refer to Table ? ) or (Refer to Figure ?), because these tables and figures have already been mentioned in the results. Also, apply your discretion to provide more depth in the discussion, tell us more of the how and why, and less of the what

Look forward to your revised manuscript. Thank you

Reviewer 2 ·

Basic reporting

The authors have included the suggested aspects. Now the introduction provides more detail on the background and significance of their work.

Experimental design

They have addressed the suggested comments.

Validity of the findings

They have addressed the suggested comments.

Additional comments

The authors addressed the suggested comments. In one figure, they include an image of the plants; however, there is no image showing the complete plants or inflorescences related to Figure 4. I believe this would be interesting, as the color variation is observed in the scale, but I wonder if this is visible in the living plants.

I have no objection to this work being approved.

---

## Round 0.3 · accepted · Accept

After a very thorough check of the revised manuscript, I am very convinced this revised manuscript is acceptable for publication. Thank you authors for finding PeerJ as your journal of choice, and looking forward to your future scholarly contributions.

Congratulations